# Selenite Foliar Application Alleviates Arsenic Uptake, Accumulation, Migration and Increases Photosynthesis of Different Upland Rice Varieties

**DOI:** 10.3390/ijerph17103621

**Published:** 2020-05-21

**Authors:** Yongzhen Ding, Xuerong Di, Gareth J. Norton, Luke Beesley, Xingxing Yin, Zulin Zhang, Suli Zhi

**Affiliations:** 1Agro-Environmental Protection Institute, Ministry of Agriculture and Rural Affairs, Tianjin 300191, China; dingyongzhen@caas.cn (Y.D.); 13126761207@163.com (X.D.); ryinxingxing@126.com (X.Y.); 2School of Biological Sciences, University of Aberdeen, Aberdeen AB24 3UU, UK; g.norton@abdn.ac.uk; 3The James Hutton Institute, Aberdeen AB15 8QH, UK; Luke.Beesley@hutton.ac.uk (L.B.); Zulin.Zhang@hutton.ac.uk (Z.Z.); 4School of Resources and Environmental Engineering, Wuhan University of Technology, Wuhan 430070, China

**Keywords:** arsenic, foliar application, photosynthetic performance, selenium, upland rice

## Abstract

This study investigates how arsenic (As) uptake, accumulation, and migration responds to selenium (Se) foliar application (0–5.0 mg × kg^−1^). Rice varieties known to accumulate low (DOURADOAGULHA) and high (SINALOAA68) concentrations of arsenic were chosen to grow on soil with different As concentrations (20.1, 65.2, 83.9 mg × kg^−1^). The results showed that Se of 1.0 mg × L^−1^ significantly alleviated As stress on upland rice grown on the As-contaminated soil. Under light (65.2 mg × kg^−1^) and moderate (83.9 mg × kg^−1^) As concentration treatments, the biomass of upland rice was increased by 23.15% and 36.46% for DOURADOAGULHA, and 46.3% and 54.9% for SINALOAA68. However, the high Se dose (5.0 mg × kg^−1^) had no significant effect on biomass and heights of upland rice compared to plants where no Se was added. Se significantly decreased As contents in stems and leaves and had different effects on As transfer coefficients for the two rice varieties: when grown on soil with low and moderate As concentrations, Se could reduce the transfer coefficient from stems to leaves, but when grown on the high As soils, this was not the case. The chlorophyll content in plants grown in soil with the moderate concentration of As could be improved by 27.4%–55.3% compared with no Se treatment. Under different As stress, the Se foliar application increased the net photosynthesis, stomatal conductance, and transpiration rate, which meant that Se could enhance the photosynthesis of rice. The intercellular CO_2_ concentration variation implied that the stomatal or non-stomatal limitations could both occur for different rice varieties under different Se application doses. In conclusion, under moderate As stress, foliar application of Se (1.0 mg × L^−1^) is recommend to overcome plant damage and As accumulation.

## 1. Introduction 

Arsenic is present in the environment due to geological sources or contamination by mining and other activities and is highly toxic to plants, animals, and humans [1,2]. Recently, arsenic pollution of water and food has become a huge public health and environmental concern in many countries, especially in Asia [3]. As-contaminated farmland soil has become the major concern as a source of As entering the food chain and therefore threatening human health [4]. In the environment, As is present in different forms, such as organic and inorganic arsenic [1,5]. The inorganic arsenic species are predominately arsenate (As(V)) and arsenite (As(III)) [6]. As the staple food for more than half of the world’s population, rice (*Oryza sativa* L.) is a major dietary source of inorganic arsenic [7,8]. Studies have shown that arsenic content in rice grains can be as high as 2.0 mg × kg^−1^ and the dominate form of As in rice grains is inorganic arsenic [9]. The value of inorganic arsenic is far above the limit determined by the Chinese Food Standards Agency (0.2 mg × kg^−1^) [10] and World Health Organization (WHO) (0.2 mg × kg^−1^) [11]. Therefore, the application of suitable techniques to reduce the concentrations of arsenic in crops is urgently needed.

Among different techniques, application of Selenium (Se) has been demonstrated to promote rice plant growth and increase it stress resistance [12]. Under As stress, the application of Se can inhibit As uptake and mitigate the toxicity of As to rice [13]. A number of possible mechanisms by which Se can alleviate heavy metals (metalloids) have been reported and proved, such as inhibiting membrane lipid peroxidation [14,15], reducing the uptake and translocation of heavy metals (metalloids) [12,15], and alleviating chloroplast damage [16]. Recently, the application of Se has been reported as an effective way to alleviate the accumulation of heavy metals (metalloids), such as Cd [10], As [17,18], Hg [19], and Sb [15]. The main forms of inorganic selenium in soil are selenate (Se(VI)) and selenite (Se(IV)). Studies have shown that the effect of Se(IV) on reducing the content of As and Cd in rice is greater than that of Se(VI) [20], and that low doses of Se are more effective than high doses of Se. Ding et al. [21] found that 0.8 mg × L^−1^ Se(IV) showed negative effects on the growth of rice plants in a hydroponic experiment. Several studies have proven that Se at low doses acts as an antioxidant or as an uptake inhibitor to counteract the toxicity of heavy metals (metalloids) in plants [10,21].

There are two main methods of inorganic Se application: (1) soil application of Se, which is the primary form of Se application, and (2) foliar application. There are a limited number of studies investigating the effects of soil Se application on plants grown in As-polluted soil/water [22]. For example, Pokhrel et al. [23] investigated the effect of selenium in soil on the toxicity and uptake of arsenic in rice and showed that the antagonism between Se and As depended on rice cultivar. Foliar application of Se has several advantages compared to soil application of Se, such as avoiding the large-scale fixation of selenite and significant leaching of selenate in the soil [24]. Foliar application of Se has been used to produce selenium-rich food [25] and reduce diseases caused by selenium deficiency. A number of studies have focused on the foliar application of Se to reduce accumulation of heavy metals such as Cd [26] and Hg [19]. However, the effects of Se foliar application on plants grown on As-polluted soil are limited. Only Ding et al. [10] reported the possibility of Se foliar application to reduce As accumulation in the edible parts of *Brassica campestris*. Since Se can not only reduce the accumulation of heavy metals (metalloids) but also improve the Se content in the food, further research is needed on major foods including rice. Moreover, different varieties of rice plants may have different responses to the foliar application with different Se doses. Therefore, the present study will provide a comprehensive examination of foliar application of Se on different rice varieties.

Plant photosynthesis is affected by diverse environmental factors such as Cd [10,27], salinity, and drought [28], leading to reduced crop yields. For example, Gao et al. [26] showed that Cd concentration significantly inhibited plant growth and photosynthesis in rice. It is generally considered that the intercellular CO_2_ concentration (Ci), stomatal conductance (Gs), the net photosynthetic rate (Pn), and transpiration rate (Tr) are the main indicators for the stomatal limitation or nonstomatal limitations [29] of photosynthesis. Chlorophyll content is also demonstrated to be a useful tool for evaluating the photosynthetic activity [30]. Up to now, some efforts have been made to enhance plant photosynthesis under different conditions; however, no information is available for the effects of Se foliar application on the photosynthesis of rice under As stress.

Rice was selected to investigate the effects of selenite foliar application on arsenic uptake, transformation, and the photosynthetic performance of plants. Low (DOURADOAGULHA, V1) and high (SINALOAA68, V2) As-accumulating upland rice cultivars were selected. The aims of this study were to: (1) investigate the effects of foliar application of Se(IV) under As stress on the upland rice growth; (2) examine the alleviating effects of Se on the As accumulation and transport in upland rice; and (3) explore the photosynthesis variation under different Se doses for different varieties of upland rice. This study will give a moderate foliar application of selenite dose for rice plants grown on As-contaminated soil.

## 2. Materials and Methods

### 2.1. Soil Pretreatment

The experimental soils (0–20 cm depth) were collected from paddy fields containing variable concentrations of As in Chenzhou, Hunan, China. According to the China Soil Environmental Quality Standard (GB 15618-2018), the soils were divided into non-polluted S1 (20.1 mg × kg^−1^), lowly polluted S2 (65.2 mg × kg^−1^), and moderately polluted S3 (83.9 mg × kg^−1^) soils. The characteristics of clay, silt, and sand of these three paddy soils were all similar. The basic physicochemical properties of the soil are shown in Table 1.

The soil was air-dried, ground, passed through a 2 mm sieve mesh, thoroughly mixed, and placed in the dark until use. Pot experiments were carried out. Each pot was filled with 3 kg soil pre-mixed with base fertilizers. The base fertilizers applied to the soil were in the following doses: N [CO(NH)_2_], 0.20 g × kg^−1^; P (KH_2_PO_4_), 0.15 g × kg^−1^; K (K_2_SO_4_), 0.20 g × kg^−1^. The soil mixed with base fertilizers was moist for a week before the rice seedlings were transplanted.

### 2.2. Plant Culture

Low (DOURADOAGULHA, V1) and high (SINALOAA68, V2) As-accumulating upland rice cultivars were selected as the experimental materials, which were provided by the Shanghai Agricultural Biogene Center. The rice seeds were sterilized with 3% NaClO for 20 min and washed three times with deionized water followed by germination on moist filter paper. The germinated seeds were grown in a greenhouse at 25 °C, with 16 h light/8 h dark photoperiod, light intensity of 350 μmol × m^−2^ × s^−1^, and relative humidity of 80%. At about 4 weeks old, the seedlings were transplanted into the experimental pots, with three plants per pot. Se as selenite (Se(IV)) was sprayed on the leaves until Se started to run off the leaves at the doses of 0 (Se0), 1.0 (Se1), and 5.0 (Se5) mg × L^−1^ at the rice tillering stage and at the booting stage. When the plants were being sprayed, plastic film was used to wrap the pot, excepting the leaf surface to prevent Se from being sprayed onto the soil. In this study, there were 9 treatments per cultivar, namely S1Se0 (without the addition of Se with a non-polluted soil), S1Se1, S1Se5, S2Se0, S2Se1, S2Se5, S3Se0, S3Se1, and S3Se5. There was a total of 18 treatments, and each treatment was replicated three times.

One week after spraying Se at the booting stage, a portable photosynthesis system (LI-6400XT) was used to measure the flag leaf photosynthesis. One day later, plant height was measured, and the plants were harvested and separated into roots and shoots. The fresh shoots were weighed and separated into stems and leaves. Some of the fresh leaf samples were immediately frozen in liquid nitrogen and stored at −80 °C to determine the chlorophyll content. The roots were immersed in an ice-cold desorption solution containing 0.5 mM Ca(NO_3_)_2_, 5 mM MES (pH 5.5), and 1 mM K_2_HPO_4_ for 15 min to remove surface-sorbed As [17]. The washed roots, stems, and remaining leaves were dried in an oven at 70 °C for 48 h to a constant weight and used for the determine of As concentration.

### 2.3. Sample Analysis

Chlorophyll was extracted from the leaves with acetone–absolute alcohol (V:V = 1:1). The digestion method of the plant samples was done following the procedure of Ding et al. [10]. In brief, 0.5 g plant samples were digested in a mixture of 8 mL HNO_3_ and 2 mL HClO_4_. The digestion process was as follows: 80 °C for 1 h, 120 °C for 1 h, and 150 °C for 1 h. The As concentrations in the digested plant samples were determined using inductively coupled plasma mass spectrometry (ICP-MS) (Agilent 7500 Series, Santa Clara, CA, USA). Standard reference material rice flour (GBW10010 (GSB-1), the Center for Standard Reference of China) was used to verify the accuracy of analysis methods (As concentration = 0.102 mg × kg^−1^).

### 2.4. Statistical Analysis

The data were subjected to analysis of variance (ANOVA) and Turky’s multi-comparisons test (*p* ≤ 0.05) using SPSS 22.0 Statistics (IBM, Armonk, NY, USA). Plots were prepared using SigmaPlot 12.5 software (Systat Software Inc., San Jose, CA, USA).

## 3. Results

### 3.1. Effect of Se on the Growth of Upland Rice under As Stress

The growth of the two selected varieties of upland rice under different arsenic stress and Se concentrations is shown in Table 2. Without the addition of Se, the growth of upland rice was significantly reduced with increasing soil arsenic concentrations; with the growth of upland rice is in the order of S1Se0 > S2Se0 > S3Se3. The shoot fresh weight and plant height of upland rice in S2Se0 treatment were 54.7% and 19.4% for DOURADOAGULHA and 65.6% and 30.7% for SINALOAA68, respectively, lower than those in the S1Se0 treatment. The shoot fresh weight and plant height of upland rice in S3Se0 treatment were 89.3% and 38.3% for DOURADOAGULHA and 92.4% and 49.0% for SINALOAA68, respectively, lower than those in the S1Se0 treatment.

Foliar application of Se had a significant effect on the rice plant biomass (*p* ≤ 0.01); however, there was no significant effect on the plant height (*p* > 0.05). The interaction of Se contents and soils containing different concentrations of arsenic (Soils * Se treatments in Table 2) had significant effects on the plant growth for different rice species (*p* ≤ 0.05). The response of plant growth with addition of foliar application of Se was variable on the different arsenic containing soils (Table 2). For example, the shoot fresh weights of DOURADOAGULHA and SINALOAA68 in S2Se1 treatment were 23.1% and 46.3%, respectively, higher than that in S2Se0 treatment. With non- and lowly As-contaminated soil, a reasonable foliar application of Se had more obvious effects on SINALOAA68 than on DOURADOAGULHA. However, the plants in foliar application of high Se (S2Se5) did not lead to significantly more biomass than the plant in the S2Se0 treatment.

### 3.2. Effect of Se on the Content of As in Upland Rice

Increasing As exposure content resulted in higher As uptake and accumulation in different parts of the plants (Figure 1). Without the addition of Se, the As contents in roots, stems, and leaves of upland rice in S2Se0 treatment were 1.8, 1.1, and 0.9 times for DOURADOAGULHA and 1.5, 1.2, and 1.0 times for SINALOAA68, respectively, of those in the S1Se0 treatment. The As contents in roots, stems, and leaves of upland rice in S3Se0 treatment were 3.1, 2.3, and 1.9 times for DOURADOAGULHA and 2.8, 2.4, and 2.1 times for SINALOAA68, respectively, of those in the S1Se0 treatment. There was a significant difference between the selected two varieties for As concentration in the roots, stems, and leaves (F = 3.126, 26.293, and 20.880, respectively; *p* ≤ 0.05). The As contents in the roots, stems, and leaves of SINALOAA68 were higher than those of DOURADOAGULHA by 33.8%–69.1%, 65.6%–148.6%, and 46.2%–110.6%, respectively.

Foliar application of Se had different effects on the As contents in different parts of the rice plants when grown on soils with different arsenic concentrations. Se did not have a significant effect on As concentration in the roots (Figure 1A). Se inhibited the accumulation of As in the stems and leaves significantly (Figure 1B,C). The S2Se1 and S2Se5 treatments reduced the As contents in stems by 14.9%–32.1% and 23.0%–38.8% in the leaves compared to the S2Se0 treatment. The S3Se1 and S3Se5 treatments reduced the As contents in stems by 22.2%–29.2% and 31.1%–43.3% in the leaves compared to the S3Se0 treatment. However, on the soil with non-arsenic (S1), Se had no significant effect on the accumulation of As in the stems and leaves.

### 3.3. Effect of Se on the Transport of As in Upland Rice

Without the addition of Se, the As transfer coefficients of roots to stems and stems to leaves of upland rice in S2Se0 treatment were lower—27.7% and 7.2% for DOURADOAGULHA and 16.5% and 6.1% for SINALOAA68, respectively—than those in the S1Se0 treatment (Table 3). The As transfer coefficients of roots to stems and stems to leaves of upland rice in S3Se0 treatment were lower—16.9% and 20.0% for DOURADOAGULHA and 11.3% and 6.9% for SINALOAA68, respectively—than those in the S1Se0 treatment. There was a significant difference between the varieties for the As transfer coefficients of roots to stems and stems to leaves (*F* = 17.771 and 7.039, respectively; *p* ≤ 0.05). The transfer coefficients of roots to stems of SINALOAA68 were higher by 9.3%–73.6% than those of DOURADOAGULHA. The transfer coefficients of stems to leaves of DOURADOAGULHA on the soil with non-polluted (S1) and lowly polluted (S2) soil were higher by 4.7%–62.7% than those of SINALOAA68. However, on the soil with high (S3) concentrations of arsenic, the transfer coefficient of stems to leaves of DOURADOAGULHA was lower by 6.5%–9.2% than those of SINALOAA68.

Foliar application of Se had different effects on the As transfer coefficients of the rice plants when grown on soils with different arsenic concentrations. On the soil with non-arsenic polluted (S1), foliar application of Se had no significant effect on the transfer coefficients of roots to stems and stems to leaves of plants except for the S1Se5 treatment, which significantly reduced the transfer coefficient of roots to stems compared to the S1Se0 treatment for DOURADOAGULHA. The transfer coefficient of stems to leaves of DOURADOAGULHA in the S2Se5 treatment was 13.4% higher than that in the S2Se0 treatment; however, compared with the S2Se0 treatment, the S2Se1 and S2Se5 treatments significantly reduced the transfer coefficients of stems to leaves of SINALOAA68 by 26.3% and 27.1%, respectively. Compared with the S3Se0 treatment, the S3Se1 and S3Se5 treatments reduced the transfer coefficients of roots to stems of upland rice by 21.7%–30.0% but had no significant effect on the transfer coefficients of stems to leaves (Table 3).

### 3.4. Effect of Se on the Chlorophyll Content of Upland Rice under As Stress

Chlorophyll content of upland rice was reduced significantly when plants were grown on soils containing higher concentrations of arsenic (Figure 2). Without the addition of Se, the chlorophyll contents of upland rice in S2Se0 and S3Se0 treatments were lowered by 45.4% and 51.6% for DOURADOAGULHA and 26.6% and 56.9% for SINALOAA68, respectively, compared to that in S1Se0 treatment.

Foliar application of Se increased the chlorophyll content of upland rice significantly in all treatments (Figure 2), compared to the relevant control in both varieties, except for S2Se1, which was not significantly different than S2Se0 for SINALOAA68, and S3Se5, which was not significantly different than S3Se0 for DOURADOAGULHA. The chlorophyll contents in plants grown in S2Se1 and S2Se5 treatments were 27.4%–55.3% higher than those in the S2Se0 treatment, respectively. Compared with the S3Se0 treatment, the S3Se1 and S3Se5 treatments increased the chlorophyll contents of rice by 6.8%–31.8%.

### 3.5. Effect of Se on Photosynthesis in Upland Rice under As Stress

The photosynthesis of the plants was significantly inhibited by the increasing concentrations of As in the soil. Without the addition of Se, the net photosynthesis (Pn) in the S2Se0 and S3Se0 treatments was significantly reduced by 24.1% and 34.3% for DOURADOAGULHA and 19.2% and 37.5% for SINALOAA68, respectively, compared with the S1Se0 treatment (Figure 3A). The stomatal conductance (Gs) was reduced significantly by 46.2% and 56.2% for DOURADOAGULHA and 33.8% and 43.3% for SINALOAA68, respectively (Figure 3B). The transpiration rate (Tr) was significantly reduced by 43.6% and 54.3% for DOURADOAGULHA and 30.8% and 53.8% for SINALOAA68, respectively (Figure 3C). However, the intercellular CO_2_ concentration (Ci) was significantly increased by 29.1% and 51.1% for DOURADOAGULHA and 14.1% and 24.6% for SINALOAA68, respectively, compared to the control plants (Figure 3D). In addition, the correlations between shoot biomass and photosynthetic parameters were analyzed in Table 4. Pn, Gs, and Tr had significant positive correlation with shoot biomass and chlorophyll content (*p* ≤ 0.05). Ci presents significant negative correlation with shoot biomass and chlorophyll content (*p* ≤ 0.01) for DOURADOAGULHA, but no significant difference at the 0.05 level was detected for SINALOAA68.

Foliar application of Se effectively alleviated the inhibition effects of As stress on photosynthesis for the rice plants grown on soil containing high concentrations of arsenic (S3). However, foliar application of Se had no significant effect on photosynthesis for the plants grown on the soil with non-arsenic (S1), except for S1Se1 and S1Se5, which had a more significant effect on Gs than S1Se0 for SINALOAA68 (Figure 3B), and S1Se5, which had a more significant effect on Pn than S1Se0 for SINALOAA68 (Figure 3A). The S2Se1 and S2Se5 treatments significantly increased the Pn, Gs, and Tr by 11.9%–21.0%, 15.2%–22.3%, and 5.4%–29.7% compared to the S2Se0 treatment, respectively, but significantly reduced the Ci by 13.0%–21.9%. The S3Se1 and S3Se5 treatments significantly increased the Pn, Gs, and Tr by 19.3%–39.4%, 10.3%–38.4% and 23.5%–55.1% compared to the S3Se0 treatment, respectively, but significantly reduced the Ci by 11.5%–20.5%.

## 4. Discussion

### 4.1. Selenium Alleviates Arsenic Stress in Upland Rice

The present study has attempted to alleviate arsenic accumulation in two varieties of upland rice plants grown in soils with different As contents. The foliar application of Se was chosen. The results showed that the growth of upland rice was inhibited when rice plants were grown in soil with increasing arsenic concentration (Table 2). An interesting observation was made in the non-arsenic polluted soils (S1), where the S1Se1 treatment had no significant effect on biomass but the S1Se5 treatment significantly reduced the biomass of the rice plants. This observation might be due to the high Se level (5 mg × L^−1^) showing a toxic effect on the growth of the upland rice which was grown in the soil with low As pollution or without As pollution; this result was not manifested in the higher arsenic soils, because the plants’ biomass was already reduced due to the existing arsenic toxicity. Han et al. [18] have reported similar results in a hydroponic experiment, with the addition of high Se (5 mg × L^−1^) inhibiting the growth of tobacco whether As was added to the solution or not.

When the plants were grown in soils containing high concentrations of As, Se increased the biomass in shoots of upland rice to some extent and reduced As contents in stems and leaves (Figure 1B,C). Foliar application of Se has been reported to facilitate plant growth under abiotic stress [31,32]. For instance, Hawrylak-Nowak et al. [33] observed that foliar application of Se significantly increased the fresh weight of lamb’s lettuce under high temperature stress, and the lamb’s lettuce showed greater thermo-tolerance. A similar result was observed in maize under salinity stress [32]. Se can promote plant growth by delaying senescence and improving water status in plants [34].

### 4.2. Selenium Inhibits Arsenic Transport in Upland Rice

In this study, foliar application of Se significantly reduced the As concentration in the stems and leaves. Similar results were obtained for root application of Se [18,35]; however, some studies have shown that application of Se has no significant effect on the contents of As in roots and stems of rice [20]. The difference may be due to the different cultivars, different application mode of Se, or different stresses. Therefore, it is necessary to further study the mechanism of Se on As uptake in rice.

Under As stress, Se significantly reduced the As transfer coefficients of roots to stems in upland rice (Table 3) in this study, which was consistent with previous reports [17]. Feng et al. [31] also observed that Se inhibited the As transfer from roots to fronds. The roots’ arsenic concentrations did not differ with different additions of Se (Figure 1A), which suggested that Se inhibited the transfer of As from roots to above-ground parts in upland rice; this may be due to the chelating reactions between Se(IV) and As in plants [20]. The decrease of the transfer coefficient of As from roots to stems probably reduced the As content in rice grain, which would decrease the potential risk of As to human [36].

### 4.3. Selenium Enhances Photosynthetic Performance of Upland Rice under As Stress

Stresses such as heavy metals, drought, and salinity can reduce plant chlorophyll content, inhibit photosynthesis and affect plant growth [14,32]. The photosynthesis intensity of plants is closely related to the chlorophyll content. In this study, the chlorophyll content reduced with the elevated arsenic concentration in soil, which has been observed by many previously studies [22]. As reduces the chlorophyll content within the plants in the following two ways: As replaced Mg^2+^ in chlorophyll and interfere with the activity of chlorophyll synthetase; and As improves the activity of chlorophyll-degrading enzyme and accelerates the decomposition of chlorophyll [14]. In this study, under As stress, foliar application of Se increased the chlorophyll content and promoted photosynthesis of upland rice, which might be because that Se reduced the accumulation of As in the leaves, thereby alleviating the damage of As to chloroplasts and improving the photosynthesis of upland rice; this was consistent with the conclusion in the study which was conducted under Cd stress [14]. Iqbal et al. [34] also observed that foliar application of Se increased the total chlorophyll contents and protected the chloroplast from oxidative damage of spring wheat under heat stress. Ashraf et al. [32] had similar results in maize under salinity stress.

For further evaluating the effects of Se on the photosynthesis of the two varieties of upland rice under different As stress, the net photosynthesis (Pn), stomatal conductance (Gs), transpiration rate (Tr), and intercellular CO_2_ concentration (Ci) were investigated first. Pn is an important indicator, which directly presents for the total photosynthesis ability of the plants. The value of Pn is affected by Gs, Tr, and Ci. Gs is a quantification parameter for the opening degree of plant mesophyll cells, which is also a key factor in the rate of carbon dioxide transfer. Tr reflects the efficiency of leaf water metabolism during photosynthesis. Usually, the variation of Gs and Tr have higher linear relation with the variation of Pn [37], which were also observed in this study (Table 4). For DOURADOAGULHA and SINALOAA68, the variation of Gs and Tr had a positive relation with the variation of Pn (*R* = 0.688 and 0.766, *p* ≤ 0.05 and 0.01, respectively). Ci means the abundance and deficiency of photosynthetic substrates in plant leaves. The variation of Ci correlated negatively with the variation of Pn for DOURADOAGULHA (R = −0.883, *p* ≤ 0.01) and the variation of Ci had no significant correlation with the variation of Pn for SINALOAA68 (Table 4). As reported, the variation of Ci can reflect that whether the change of photosynthesis is caused by stomatal factors or non-stomatal factors [28]. In our study, with the increasing of As stress, the Pn, Gs, and Tr were all decreased, but Ci was increased without addition of Se (Figure 3). This indicated that the main causes of the decrease of photosynthesis were non-stomatal factors. When the foliar application of Se was used, the values of Pn, Gs, and Tr were increased, which means that Se could mitigate the damage of the plants caused by As and enhance photosynthetic performance of upland rice under As stress. However, for different varieties of plants (DOURADOAGULHA and SINALOAA68), the effects of Se had a little difference. For example, for DOURADOAGULHA, the values of Pn in the treatments with 5 mg × kg^−1^ Se were higher than those in the treatments with 1 mg × kg^−1^ Se, but for SINALOAA68, the values of Pn in the treatments with 5 mg × kg^−1^ Se were lower than those in the treatments with 1 mg × kg^−1^ Se (Figure 3A). The same trend was observed for the values of Tr (Figure 3C). This means that different plants may have different tolerance to Se; 5 mg × kg^−1^ of Se may have some negative effects on the photosynthetic performance of plants. Wang et al. [38] found that high level of Se significantly damaged the photosynthetic apparatus and inhibited the photosynthesis of rice. Tang et al. [19] also pointed out that, although Se could reduce the Hg accumulation in rice plant, special attention should be paid to Se doses considering that Se could be a toxic at higher doses. Under different application of Se, for DOURADOAGULHA, the values of Ci were decreased by Se, but the values of Ci were also increasing as the As stress increased (Figure 3D). This means that, for DOURADOAGULHA, the decrease of photosynthesis was mainly caused by non-stomatal factors; this showed that stomatal limitation is the main factor for the variation of Ci during the Se application. However, for SINALOAA68, although the values of Ci were increased by As stress, the values of Ci were turning from increasing to decreasing with the As stress increasing (Figure 3D). This implied that photosynthesis might be influenced by both stomatal factors and non-stomatal factors for SINALOAA68. During a non-stomatal limitation process, CO_2_ could not be effectively utilized by the photosynthesis. The accumulation of CO_2_ concentration damaged the normal operations of the photosynthetic process. This nonstomatal limitation process was also reported in other studies. For example, Li et al. [28] investigated the photosynthetic properties of *Lycium ruthenicum* Murr under salinity and drought, and observed typical non-stomatal limitations under high salinity and drought stress. Similar results also were reported by Chu et al. [29], who showed that the reduction in photosynthesis of *Schima superba* young plants was affected mainly by non-stomatal limitations under Cd stress. Above all, foliar application of an appropriate dose of Se can help to slow down the harmful effects of As stress, but excessive use of Se can also produce toxic effects on the plants. Plant varieties are also important; different crops have different tolerances to Se and As. The results manifested might be helpful to manage the rice in As-contaminated regions.

## 5. Conclusions

Increasing soil arsenic concentrations significantly inhibited the growth of upland rice. Low concentrations of Se (1.0 mg × L^−1^) effectively alleviated As stress; however, high concentrations of Se (5.0 mg × L^−1^) had no significant effect. Foliar application of Se significantly reduced the As uptake in stems and leaves, and the transfer coefficient of As from roots to stems. In addition, Se application increased the chlorophyll content and promoted photosynthesis of upland rice. The difference in low and high As-accumulating upland rice was that the As uptake and transfer coefficient of As from roots to stems in the high accumulating variety were much higher than those in the low accumulating variety. In conclusion, in moderately contaminated arsenic soils, foliar application of Se (1.0 mg × L^−1^) is recommended for overcoming the reduction in plant biomass and reduced arsenic accumulation in stems and leaves.

## Figures and Tables

**Figure 1 ijerph-17-03621-f001:**
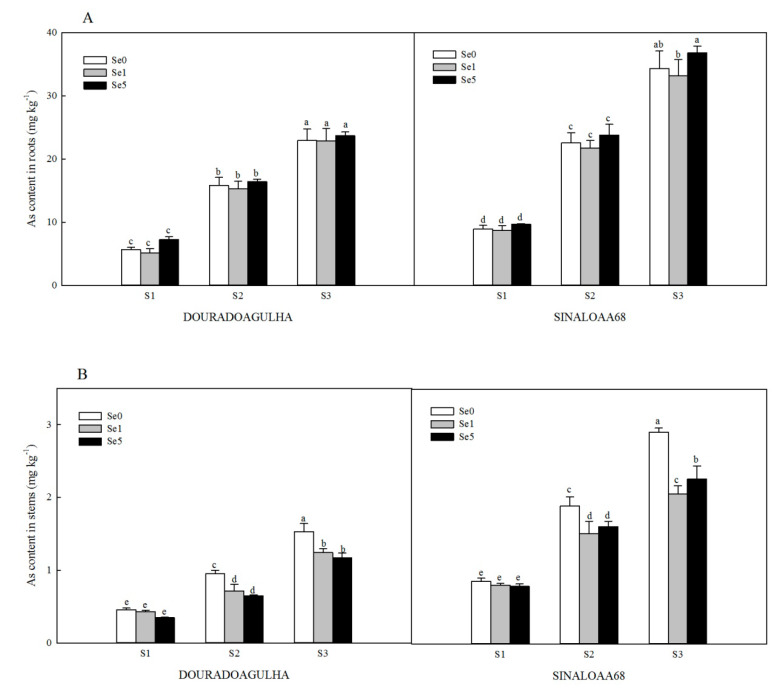
Effect of Se on As content in roots (**A**), stems (**B**), and leaves (**C**) of DOURADOAGULHA and SINALOAA68 grown on soils with different arsenic concentrations (S1 = 20.1 mg × kg^−1^, S2 = 65.2 mg × kg^−1^, S3 = 83.9 mg × kg^−1^). Different letters above the bars indicate significant difference between treatments at *p* ≤ 0.05.

**Figure 2 ijerph-17-03621-f002:**
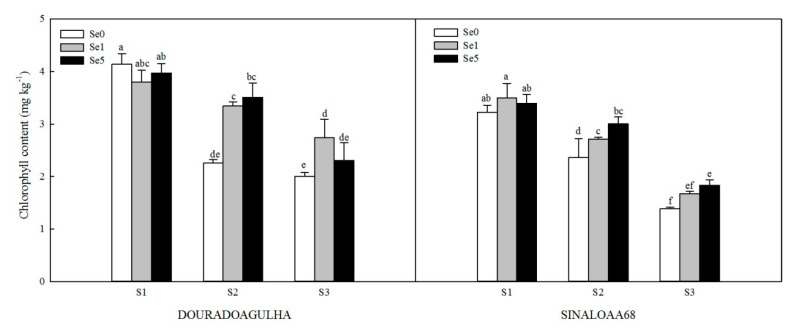
Effect of Se on chlorophyll content of DOURADOAGULHA and SINALOAA68 grown on soils with different arsenic concentrations (S1 = 20.1 mg × kg^−1^, S2 = 65.2 mg × kg^−1^, S3 = 83.9 mg × kg^−1^). Different letters above bars indicate significant difference between treatments at *p* ≤ 0.05.

**Figure 3 ijerph-17-03621-f003:**
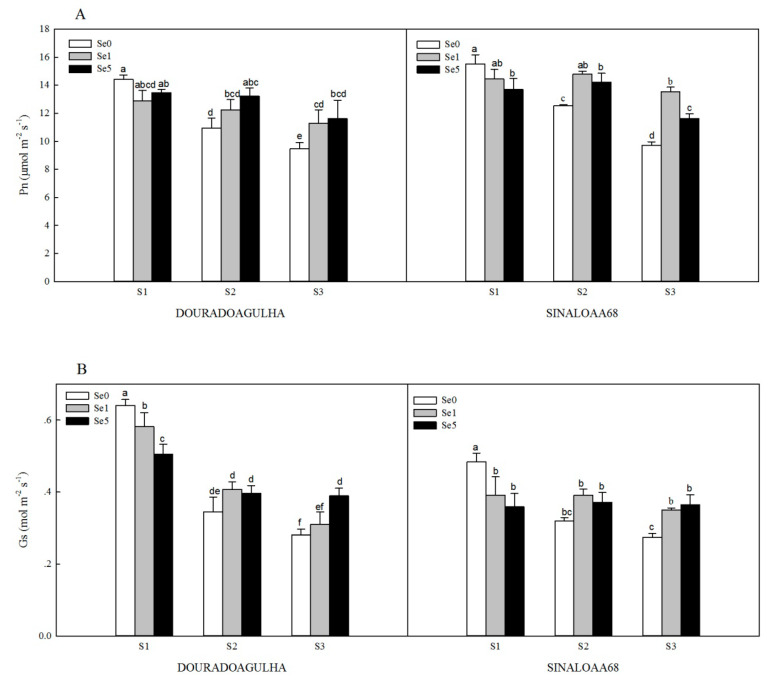
Effect of Se on net photosynthesis (Pn) (**A**), stomatal conductance (Gs) (**B**), transpiration rate (Tr) (**C**), and intercellular CO_2_ concentration (Ci) (**D**) of DOURADOAGULHA and SINALOAA68 grown on soils with different arsenic concentrations (S1 = 20.1 mg × kg^−1^, S2 = 65.2 mg × kg^−1^, S3 = 83.9 mg × kg^−1^). Different letters above bars indicate significant difference between treatments at *p* ≤ 0.05.

**Table 1 ijerph-17-03621-t001:** Physicochemical characteristics of the soil.

Soils	pH	CEC ^1^ (cmol × L^−1^)	Organic Matter (g × kg^−1^)	Total As (mg × kg^−1^)	Total Se (mg × kg^−1^)	Total N (g × kg^−1^)	Total K (g × kg^−1^)	Total P (g × kg^−1^)	Available P (mg × kg^−1^)
S1	6.65	12.47	24.3	20.1	0.65	1.69	16.36	0.49	11.63
S2	7.56	13.83	45.6	65.2	0.91	2.60	17.51	1.12	64.59
S3	7.76	13.68	38.9	83.9	0.86	2.26	18.77	1.25	50.29

^1^ CEC is cation-exchange capacity.

**Table 2 ijerph-17-03621-t002:** Effect of Se on the biomass in shoots and height of upland rice (DOURADOAGULHA and SINALOAA68) grown on soils containing different concentrations of arsenic (S1 = 20.1 mg × kg^−1^, S2 = 65.2 mg × kg^−1^, S3 = 83.9 mg × kg^−1^).

Soils	Se Treatments (mg × L^−1^)	DOURADOAGULHA		SINALOAA68	
		Shoot Fresh Weight (g × plant^−1^)	Plant Height (cm)	Shoot Fresh Weight (g × plant^−1^)	Plant Height (cm)
S1	Se0	14.37 ± 0.66 ^a^	83.87 ± 4.44 ^a^	24.01 ± 1.98 ^a^	106.47 ± 11.48 ^a^
	Se1	15.16 ± 0.30 ^a^	85.67 ± 3.26 ^a^	23.19 ± 1.52 ^a^	98.33 ± 2.08 ^a^
	Se5	11.11 ± 0.37 ^b^	87.33 ± 0.29 ^a^	18.05 ± 0.25 ^b^	89.00 ± 2.65 ^b^
S2	Se0	6.09 ± 0.61 ^d^	67.67 ± 3.79 ^b^	7.43 ± 0.08 ^d^	65.67 ± 2.52 ^c,d^
	Se1	7.50 ± 0.57 ^c^	69.33 ± 3.51 ^b^	10.87 ± 1.26 ^c^	72.67 ± 2.08 ^c^
	Se5	6.18 ± 0.35 ^d^	69.67 ± 5.51 ^b^	7.89 ± 0.57 ^d^	68.33 ± 3.51 ^c,d^
S3	Se0	1.44 ± 0.00 ^e^	58.00 ± 6.08 ^c^	1.64 ± 0.13 ^e^	54.33 ± 6.43 ^e^
	Se1	1.97 ± 0.18 ^e^	67.67 ± 2.08 ^b^	2.54 ± 0.03 ^e^	60.00 ± 2.00 ^d,e^
	Se5	1.29 ± 0.24 ^e^	53.33 ± 2.08 ^c^	2.62 ± 0.23 ^e^	36.00 ± 4.58 ^f^
Significance level (*p* values)					
Soils		**	**	**	**
Se treatments		**	NS	**	NS
Soils * Se treatments		**	*	**	**

Values represent means ± standard deviations of the three replicates per treatment. Different letters (^a^ to ^f^) indicate significant difference between treatments at *p* ≤ 0.05. * and ** indicate significant differences at *p* ≤ 0.05 and 0.01, respectively. NS means no significant difference at the 0.05 level of probability.

**Table 3 ijerph-17-03621-t003:** Effects of Se on the As transport coefficients of upland rice (DOURADOAGULHA and SINALOAA68) grown on soils containing different concentrations of arsenic (S1 = 20.1 mg × kg^−1^, S2 = 65.2 mg × kg^−1^, S3 = 83.9 mg × kg^−1^).

Soils	Se Treatments (mg × L^−1^)	DOURADOAGULHA		SINALOAA68	
		Stem/Root	Leaf/Stem	Stem/Root	Leaf/Stem
S1	Se0	0.083 ± 0.008 ^a^	0.863 ± 0.049 ^a,b^	0.097 ± 0.007 ^a^	0.815 ± 0.015 ^a^
	Se1	0.085 ± 0.015 ^a^	0.864 ± 0.145 ^a,b^	0.094 ± 0.007 ^a^	0.713 ± 0.076 ^a,b^
	Se5	0.046 ± 0.000 ^c^	0.969 ± 0.020 ^a^	0.081 ± 0.003 ^a,b,c,d^	0.750 ± 0.014 ^a^
S2	Se0	0.060 ± 0.002 ^b,c^	0.801 ± 0.024 ^a,b,c^	0.084 ± 0.009 ^a,b,c^	0.765 ± 0.051 ^a^
	Se1	0.047 ± 0.009 ^c^	0.811 ± 0.051 ^a,b,c^	0.070 ± 0.012 ^b,c,d^	0.564 ± 0.083 ^c^
	Se5	0.040 ± 0.000 ^c^	0.908 ± 0.042 ^a^	0.068 ± 0.004 ^b,c,d^	0.558 ± 0.067 ^c^
S3	Se0	0.069 ± 0.002 ^a,b^	0.690 ± 0.128 ^b,c,d^	0.086 ± 0.008 ^a,b^	0.759 ± 0.048 ^a^
	Se1	0.054 ± 0.003 ^b,c^	0.552 ± 0.054 ^d^	0.064 ± 0.002 ^c,d^	0.607 ± 0.002 ^b,c^
	Se5	0.049 ± 0.004 ^c^	0.632 ± 0.083 ^c,d^	0.062 ± 0.007 ^d^	0.676 ± 0.094 ^a,b,c^
Significance level (*p* values)					
Soils		*	**	**	*
Se treatments		*	NS	**	**
Soils * Se treatments		*	NS	NS	NS

Values represent means ± standard deviations of the three replicates per treatment. The letters (^a^, ^b^, ^c^ and ^d^) indicate significant difference between treatments at *p* ≤ 0.05. * and ** indicate significant differences at *p* ≤ 0.05 and 0.01, respectively. NS means no significant difference at the 0.05 level of probability.

**Table 4 ijerph-17-03621-t004:** Correlations between shoot biomass and photosynthetic parameters of upland rice (DOURADOAGULHA and SINALOAA68).

Parameters	DOURADOAGULHA					SINALOAA68				
	Chlorophyll	Pn	Gs	Tr	Ci	Chlorophyll	Pn	Gs	Tr	Ci
Shoot Fresh Weight	0.932 **	0.645 *	0.905 **	0.829 **	−0.849 **	0.895 **	0.752 *	0.674 *	0.916 **	NS
Chlorophyll		0.810 **	0.873 **	0.832 **	−0.939 **		0.821 **	0.695 *	0.895 **	NS
Pn			0.688 *	0.766 **	−0.883 **			0.826 **	0.904 **	NS
Gs				0.914 **	−0.844 **				0.812 **	NS
Tr					−0.871 **					NS

Values represent Pearson Correlation Coefficient. * and ** indicate significant differences at *p* ≤ 0.05 and 0.01, respectively. NS means no significant difference at the 0.05 level of probability.

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
