# Peer review of "Selenite Foliar Application Alleviates Arsenic Uptake, Accumulation, Migration and Increases Photosynthesis of Different Upland Rice Varieties"

_ijerph, 2020, doi:10.3390/ijerph17103621_

Round 1

Reviewer 1 Report

good clear study, effective experimental design. some minor comments /modifications:

  1. other nutrients - differences in eg. P availabilty between soils is quite significant, other soil nutrients similar?
  2. whilst 2mm mesh used to prepare experimental medium, and texture consistent across samples, is the difference in pollution associated with any matrix difference? all soils contaminated from same source???
  3. Check manuscript - the term "heavy metals" should be changed as you refer to a range of metals metalloids, and some elements which are and other which are not "heavy metals"
  4. comment on other supplements e.g. Si which might have parallel effects, also benchmark Se additions to toxicity effects
  5. I assume you didnt measure Se in plant material because of high external dose.... but other elements associated with As/Se cycling e.g. Fe might have been useful? comment perhaps ?

Author Response

Reviewer #1

Comment 1: other nutrients - differences in eg. P availabilty between soils is quite significant, other soil nutrients similar?

Response: Thanks. In Table 1, we have added the total K content, and changed the order of the physico-chemical characteristics. We also have changed the unit of total N and P. We believe that Table 1 includes the main soil nutritional parameters.

Comment 2: whilst 2mm mesh used to prepare experimental medium, and texture consistent across samples, is the difference in pollution associated with any matrix difference? all soils contaminated from same source???

Response: The three paddy soils were collected from the same area, which was contaminated from historical sewage irrigation because of metal smelting. The difference in soil pollution was caused by the amount of sewage irrigation. And the characteristics of clay, silt and sand of these three paddy soils were all similar, which is represented in the first paragraph of section 2.1.

Comment 3: Check manuscript - the term "heavy metals" should be changed as you refer to a range of metals metalloids, and some elements which are and other which are not "heavy metals".

Response: Thanks. Changes have been made according to the suggestion, such as “heavy metals (metalloids)”.

Comment 4: comment on other supplements e.g. Si which might have parallel effects, also benchmark Se additions to toxicity effects.

Response: I agree your opinion. We have added some relevant reports at the end of the second paragraph of introduction, as follows:

“…And low doses of Se are more effective than high doses of Se. Ding et al. [21] found that 0.8 mg·L−1 Se(IV) showed negative effects on the growth of rice plants in a hydroponic experiment. Several studies have proven that Se at low doses act as an antioxidant or as an uptake inhibitor to counteract the toxicity of heavy metals (metalloids) in plants [10, 21].…”

Sorry, we didn’t analyze Si, because we think it is not our topic in this manuscript.

Comment 5: I assume you didnt measure Se in plant material because of high external dose.... but other elements associated with As/Se cycling e.g. Fe might have been useful? comment perhaps ?

Response: Your opinion is very useful and important. But it is a shame that we had neglected the measurement and analysis of Se and other elements. Undoubtedly, we will consider your suggestions in future researches. Thank you very much.

Reviewer 2 Report

General Comments:

This manuscript deals with "reducing arsenic uptake by using selenite foliar application". It is an interesting topic, but needs some revisions before to be considered for publication.

Rice (Oryza sativa L.) provides food for more than three billion people. Approximately 90% of rice production and consumption is reported in Asia. However, rice consumption can pose problems because of the arsenic (As) accumulation in rice and thus serves a vital source of As exposure in humans (Abedi and Mojiri, 2020).
Abedi, T.; Mojiri, A.; Arsenic Uptake and Accumulation Mechanisms in Rice Species. Plants, 2020, 9, 129. doi: 10.3390/plants9020129

1. Abstract:

1.1. Start abstract with motivational statement.

1.2. Page 1, Line 19; "...results showed that,..." Delete ","

1.3. Write keywords alphabetically.

2. Introduction:

2.1. Page 2, Line 43; "...As is present in a number of forms..." Should be corrected as "As is present in different forms (such as: organic and inorganic arsenic)….".

3. Materials and Methods:

3.1. Page 1, Line 104; "non-polluted S1 (20.1 mg·kg-1)…" 20.1 mg/Kg of As in soil has been considered as non-polluted?!!!!! Please check the unite in Chinese Standard!!! 

3.2. Page 3, Line 127; "....S1Se1, S1Se5, S2Se0, S2Se1, 127 S2Se5, S3Se0, S3Se1, S3Se5." Se concentration of each treatment?

Author Response

Reviewer #2

This manuscript deals with "reducing arsenic uptake by using selenite foliar application". It is an interesting topic, but needs some revisions before to be considered for publication.

Rice (Oryza sativa L.) provides food for more than three billion people. Approximately 90% of rice production and consumption is reported in Asia. However, rice consumption can pose problems because of the arsenic (As) accumulation in rice and thus serves a vital source of As exposure in humans (Abedi and Mojiri, 2020).

Abedi, T.; Mojiri, A.; Arsenic Uptake and Accumulation Mechanisms in Rice Species. Plants, 2020, 9, 129. doi: 10.3390/plants9020129

Response: Thanks for your suggestion. We have added this as reference [8].

  1. Abstract:

1.1. Start abstract with motivational statement.

1.2. Page 1, Line 19; "...results showed that,..." Delete ","

Response: Thanks you very much. We have deleted it.

1.3. Write keywords alphabetically.

Response: Thanks. We have changed according to this suggestion.

  1. Introduction:

2.1. Page 2, Line 43; "...As is present in a number of forms..." Should be corrected as "As is present in different forms (such as: organic and inorganic arsenic)….".

Response: Thanks you very much. We have changed according to your opinion.

  1. Materials and Methods:

3.1. Page 1, Line 104; "non-polluted S1 (20.1 mg·kg-1)…" 20.1 mg/Kg of As in soil has been considered as non-polluted?!!!!! Please check the unite in Chinese Standard!!!

Response: Thank you very much. We have checked again. In Chinese Standard (GB 15618-2018), when pH of paddy soil is between 6.5 and 7.5, As content exceeding 25 mg·kg-1 is considered as “polluted”.

3.2. Page 3, Line 127; "....S1Se1, S1Se5, S2Se0, S2Se1, 127 S2Se5, S3Se0, S3Se1, S3Se5." Se concentration of each treatment?

Response: Thank you. Se concentration of each treatment was stated in the previous sentence “…at the doses of 0 (Se0), 1.0 (Se1), 5.0 (Se5) mg·L-1” in the same paragraph.

Reviewer 3 Report

The manuscript is very interesting, well written, discussion is good.

Title – ok. correct

Abstract – correct, suitable

  1. 34 - Please add dose of selenium, maybe 1 mg/kg ?

Key words – acceptable

Introduction

  1. 39 - Please give source pollution As, from Industry ?

L 49 - for example - remediation of As by plants ? or chemical and physical methods ?

  1. 52 - Are You sure ? Se in amino acid ?

L 53 – Maybe add species of plant, which Se tolerate and like ? For examples: Astragalus glycyphyllos L

I have suggest add paper about impact of Cd on plant, photosynthetic, for examples:  

Bączek-Kwinta R., Juzoń K., Borek M., Antonkiewicz J. 2019. Photosynthetic response of cabbage in cadmium-spiked soil. Photosynthetica, 57, 3, 731-739. DOI: 10.32615/ps.2019.070

The aims of this study was present correct.

Material and methods

In table 1 Please change unit from mg/kg to g/kg for total P and total N

  1. 105 - Please give systematic of soil according WRB or China ...

Pot experiment was correct according standards method,

L, 137 – How long time

  1. 158 - Please give for example 54.7% no 54.72% and see next numbers .....
  2. 179 - maybe content ? no uptake, because uptake is physiological process ....

Why not determined selenium in plants – rice ?

Discussion

In the future I suggest the determination of amino acids binding Selenium in rice

Conclusion

  1. 411 - For example please add dose of selenium

Author Response

Reviewer #3

The manuscript is very interesting, well written, discussion is good.

Title – ok. correct

Abstract – correct, suitable

L 34 - Please add dose of selenium, maybe 1 mg/kg ?

Response: Thanks. We have added.

Key words – acceptable

Introduction

L 39 - Please give source pollution As, from Industry?

Response: We have changed the sentence and added some references, as “…Arsenic is present in the environment due to geological sources or contamination by mining and other activities…”.

L 49 - for example - remediation of As by plants ? or chemical and physical methods ?

Response: Yes. There are really some ways to reduce As pollution, such as water managements, soil conditioners etc. Here the “suitable techniques” is the purpose of this article. The next paragraph will introduce the use of Se as an effective way to reduce metal pollution. We revised the next sentence as follow: “Among different techniques, application of Selenium (Se) has been demonstrated to promote rice plant growth and increase it stress resistance…”.

L 52 - Are You sure ? Se in amino acid ?

Response: we are not sure. To avoid confusion, we deleted this sentence. Because we think this sentence is not important.

L 53 – Maybe add species of plant, which Se tolerate and like? For examples: Astragalus glycyphyllos L.

Response: Thanks. We have made a correction and added rice as the species of the plant.

I have suggest add paper about impact of Cd on plant, photosynthetic, for examples: 

Bączek-Kwinta R., Juzoń K., Borek M., Antonkiewicz J. 2019. Photosynthetic response of cabbage in cadmium-spiked soil. Photosynthetica, 57, 3, 731-739. DOI: 10.32615/ps.2019.070

Response: Thanks for your suggestion. We have added this reference in the first sentence of the fourth paragraph in introduction.

The aims of this study was present correct.

Material and methods

In table 1 Please change unit from mg/kg to g/kg for total P and total N

Response: Thanks for your suggestion. We have changed “mg/kg” to “g/kg”.

L 105 - Please give systematic of soil according WRB or China ...

Pot experiment was correct according standards method.

Response: Thank you. We have changed the soil to paddy soils.

L, 137 – How long time

Response: Thank you. We have added the time “for 48 h”.

L 158 - Please give for example 54.7% no 54.72% and see next numbers .....

Response: Thanks you very much. We have changed the numbers according to this suggestion.

L 179 - maybe content ? no uptake, because uptake is physiological process ....

Response: Thanks. We have changed “uptake” to “content”.

Why not determined selenium in plants – rice ?

Response: Your opinion is very important. It is a shame that we had neglected the measurement of Se. We will consider your suggestions in future researches. Thank you very much.

Discussion

In the future I suggest the determination of amino acids binding Selenium in rice

Response: Thank you very much for your suggestion. It is really important. It is a shame that we had neglected the measurement of Se. We will consider your suggestions in future researches. Thank you very much.

Conclusion

L 411 - For example please add dose of selenium

Response: Thank you very much. We have added the dose of Se in the conclusion.

Round 2

Reviewer 2 Report

Reviewers’ comments have been addressed